# PRL/microRNA-183/IRS1 Pathway Regulates Milk Fat Metabolism in Cow Mammary Epithelial Cells

**DOI:** 10.3390/genes11020196

**Published:** 2020-02-13

**Authors:** Peixin Jiao, Yuan Yuan, Meimei Zhang, Youran Sun, Chuanzi Wei, Xiaolai Xie, Yonggen Zhang, Sutian Wang, Zhi Chen, Xiaolong Wang

**Affiliations:** 1College of Animal Science and Technology, Northeast Agricultural University, Harbin 150030, China; jiaopeixin@163.com (P.J.); 15546120650@163.com (M.Z.); syr1000@126.com (Y.S.); chuanziwei1225@163.com (C.W.); xlxie123@126.com (X.X.); zhangyonggen@sina.com (Y.Z.); 2School of Nursing, Yangzhou University, Yangzhou 225009, China; 006707@yzu.edu.cn; 3State Key Laboratory of Livestock and Poultry Breeding, Institute of Animal Science, Guangdong Academy of Agricultural Sciences, Guangzhou 510640, China; wstlyt@126.com; 4College of Animal Science and Technology, Yangzhou University, Yangzhou 225009, China; wangxl@yzu.edu.cn

**Keywords:** PRL, miR-183, milk fat metabolism, IRS1, dairy cows

## Abstract

The aim of the study was to understand the internal relationship between milk quality and lipid metabolism in cow mammary glands. A serial of studies was conducted to assess the molecular mechanism of PRL/microRNA-183/IRS1 (Insulin receptor substrate) pathway, which regulates milk fat metabolism in dairy cows. microRNA-183 (miR-183) was overexpressed and inhibited in cow mammary epithelial cells (CMECs), and its function was detected. The function of miR-183 in inhibiting milk fat metabolism was clarified by triglycerides (TAG), cholesterol and marker genes. There is a CpG island in the 5′-flanking promoter area of miR-183, which may inhibit the expression of miR-183 after methylation. Our results showed that prolactin (PRL) inhibited the expression of miR-183 by methylating the 5′ terminal CpG island of miR-183. The upstream regulation of PRL on miR-183 was demonstrated, and construction of the lipid metabolism regulation network of microRNA-183 and target gene IRS1 was performed. These results reveal the molecular mechanism of PRL/miR-183/IRS1 pathway regulating milk fat metabolism in dairy cows, thus providing an experimental basis for the improvement of milk quality.

## 1. Introduction

Researchers have focused on improving the milk yield and quality of dairy cows for several decades. Recently, research focusing on topics from ruminant nutrition to physiological perspectives has been performed with modern molecular techniques, such as microarrays or RNA sequencing. The mammary gland has been extensively studied to illustrate the mechanism of milk synthesis and secretion [1]. Due to the genetic and functional similarities of prolactin (PRL), growth hormone and placental PRL, researchers believe that they evolved from the same progenitor gene [2]. PRL plays an indispensable role in lactation [3]; however, the regulatory genes or mechanism of PRL is still not clarified. In this study, the results showed that PRL downregulated the expression of microRNA-183 (miR-183).

The growth and development of mammary gland tissue and lactation are complex processes that require precise regulation from the body [4,5]. MicroRNAs (miRNAs) regulate gene expression after transcription and participate in major aspects of the life activities of the organism, such as organ development [6,7], cell differentiation [8], cell proliferation and apoptosis [9], cell lipid metabolism [10], sugar metabolism [11] and protein metabolism [12]. In lactation physiology, numerous genes are associated with the growth, development and lactation of mammary gland tissue [13,14]. Previous results showed that the expression of 96 and 133 miRNAs was highly related to mammary and adipose tissues, respectively. Inherently, the copy numbers of miRNAs co-expressed in these two tissues were different [15]. This finding may suggest that miRNAs are involved in fat metabolism in mammary tissues. Previous studies on the regulation of miRNAs in mammary tissue lipid metabolism mainly focused on the analysis of miRNA expression profiles in mammary at different lactation stages by sequencing technology. For example, Avril-Sassen et al. 2009 [16] sequenced, analyzed and compared the expression profiles of miRNAs from the mammary tissues of mice in different lactation stages. Results showed that some miRNA expression levels (e.g., miR-1, miR-200, miR-126 and miR-146) were significantly different at the early and middle lactation stages, indicating that these differentially expressed miRNAs were related to mammary lactation and lactase metabolism. Many miRNA candidates by sequencing animal mammary gland tissue were obtained in previous studies, but there is a lack of in-depth systematic functional verification and mechanistic exploration of these miRNAs, and the roles and molecular mechanisms of some important candidate miRNAs are still unclear. Therefore, the main purpose of this study was to select differentially expressed miR-183 under PRL treatment and explore the function of miR-183 in isolated mammary epithelial cells.

## 2. Material and Methods

### 2.1. Cell Culture and Treatment

The basic culture medium of the primary mammary epithelial cells of Holstein cows was composed of DMEM/F12, 10% fetal bovine serum and various cytokines (e.g., 5 g/mL bovine insulin, 10 kU/L cyanine/streptomycin, etc.). Mammary gland epithelial cells recovered at 37 °C, 5% CO_2_ and suitable humidity for 48 h and were treated with PRL at the following concentrations (0 g/mL, 0.5 g/mL, 1 g/mL, 2 g/mL, and 4 g/mL), 5-aza or PRL +5-aza-dC (10 µM). Cells were collected after 48 h for subsequent evaluation. Cow mammary epithelial cells were cultured with PRL for 48 h. Each treatment contained triplicate samples.

### 2.2. Extraction and Quality Determination of Total RNA

Total RNA was extracted from mammary tissue and cells by the TRIzol method according to the instructions of the RNA extraction kit (Invitrogen Corp., Carlsbad, CA, USA) [17]. The integrity of total RNA was detected by agarose gel, and the concentration was determined by a nucleic acid protein analyzer. The qualified RNA samples were stored at −80 °C for later use.

### 2.3. Triglyceride (TAG) Content and Cholesterol Detection

A six-well cell culture plate was used for cell cultivation, and three replicates were set for each sample. The samples were washed with PBS, and then, pyrolysis liquid (150 μL) was added after removing PBS. Cells and pyrolysis liquid were collected in EP tubes (1.5 mL) after 15 min at 4 °C. The cells were then lysed on ice and centrifuged. Ten microliters of supernatant samples were collected for BCA protein detection. TAG and cholesterol kits (Loogen, Beijing, China) were used for TAG and cholesterol content detection according to the method introduced by Xu et al. in 2018 [18]. 

### 2.4. Expression Level Detection in Fluorescence Quantitative PCR

Detection of miRNA expression: RNA was extracted from transfected cells, and miRNA in total RNA was reverse transcribed by following the instructions of a miRNA reverse transcription kit (TaKaRa, Osaka, Japan) [19]. Quantitative primers were designed according to the miR-183 sequence based on the miRbase database (http://www.mirbase.org) (Appendix A). Detection of mRNA expression: reverse transcription was performed according to the instructions of the reverse transcription kit (TaKaRa, Osaka, Japan) [19]. Fluorescence quantitative PCR primers were designed according to the sequences of bovine IRS1, SCD (Stearoyl-CoA desaturase), ABCG1 (ATP-binding cassette sub-family G member 1), FASN (Fatty acid synthase), ABCA1 (ATP binding cassette subfamily A member 1), HSL (Hormone sensitive lipase), CPT1 (Carnitine palmitoyltransferase 1) and the internal reference gene UXT (Ubiquitously expressed transcript) from *NCBI* (Appendix A).

### 2.5. Transfection of miR-183 Mimics and Inhibitor in Primary Cow Mammary Epithelial Cells (CMECs)

Preliminary results showed that the optimal transfection concentration of miR-183 mimics was 100 nM and that the miR-183 inhibitor was 300 nM. Specific targeted DNA methyltransferase, which is the amino terminal siRNA, was designed and synthesized. Each sample was set for three replicates (Appendix A).

### 2.6. Western Blot Analysis

Proteins were extracted from the transfected cells after 48 h of transfection with miR-183 mimic and inhibitor. The OD data were measured at 570 nm with a microplate micrograph, and a standard curve was used to calculate the protein concentration of the samples. The test samples were loaded with 20 μL protein samples and 10 μL protein marker per well on separated and concentrated gels. Proteins were separated by SDS-PAGE and transferred to a nitrocellulose membrane (Millipore, Boston, MI, USA). Then, the samples were probed with primary monoclonal rabbit anti-IRS1 (Cell Signaling, #2382) and monoclonal mouse antibodies plus anti-β-actin antibody (Proteintech Group, 66009-1-IG, Wuhan, China). The reagents were transferred to the membrane and diluted. The samples were incubated with good resistance at 4 °C overnight. The first diluent was recovered, and the TBST membrane was washed on a shaker 3 times (5 min each time), followed by the secondary antibody. Signals were detected by the chemiluminescent ECL Western blot system (Pierce, Rockford, IL, USA).

### 2.7. Verification of the Dual Luciferase Reporter Gene Targeting IRS1 by miR-183

miR-183 was selected for further analysis of the relationship between miR-183 and its target genes. Online software TargetScan v6.2 and the miRNA function analysis software DAVID (Functional Annotation Bioinformatics Microarray Analysis) (https://david.ncifcrf.gov/summary) were used. We separately constructed IRS1 gene 3′-UTR wild type and mutant vectors (Appendix A), and then vectors were transferred into luciferase report vector to test activity. To determine whether miR-183 targets these sites directly, the 3′-UTR fragment of the IRS1 gene, which contains the target site of miR-183, was synthesized. This fragment (including the target site of miR-183) was transferred to the psi-check2 vector for cloning and identification. Seventy-five microlitres of PBS and luciferase substrate were added to each well of plates, and then, the fluorescence value of luciferase was determined with a microplate assay. Seventy-five microlitres of stop reagent was added to plates, and Renilla luciferase fluorescence was determined by a microplate assay after 10 min of incubation in the dark. The fluorescence ratio was calculated based on the value before and after.

### 2.8. Bisulphite Sequencing PCR (BSP) of miR-183

The cells were pretreated with PRL (including different times and concentrations) and inoculated in 6-well plates. An Axygen genomic DNA purification and extraction kit (Qiagen, Shanghai, China) was used for cell genomic DNA extraction. Four hundred and fifty ng genomic DNA were purified and recovered after bisulphite modification with the EZ DNA Methylation-Gold TM kit (Qiagen, Shanghai, China). Genomic DNA modified with bisulphite was used as the template, and BSP was used as a primer to amplify the methylated fragment of the miR-183 5′ promoter. The PCR product was cloned into the pmd-19t vector, and the Top10 competent cells were transformed [20]. The mono-clones were collected and sent to Shanghai Yingjun Biotechnology Co., Ltd. for sequencing (Yingjun, Shanghai, China).

### 2.9. Statistical Analysis

In order to visually exhibit the stability of each candidate gene were used to process the raw Ct values obtained by qRT-PCR. Data for the geNorm and NormFinder algorithms had to be processed by the formula: 2^−ΔCt^ (ΔCt = each corresponding Ct value—the minimum Ct value). Then, by importing 2^−ΔCt^ values into the programs, the stability parameters of each gene could be obtained. On the other hand, the best keeper was used for unification. BestKeeper (UXT) uses a different mechanism to choose the most stable genes. It can intuitively select the internal reference genes more accurately under different external environments. The results also represent relative quantities. The relative concentrations of triglycerides and cholesterol were corrected by protein concentration (BCA protein detection) after detecting their absolute values. SPSS18.0 software was used for statistical analysis, mean standard deviation was used for continuous variables, and one-way ANOVA (Analysis of Variance) was used as a model of analysis. Significance was declared at *p* < 0.05. GraphPad Prism v5.0 was used to draw the figures; * *p* < 0.05; ** *p* < 0.01; *** *p* < 0.001. All experiments were duplicated and repeated three times. Values are presented as means ± standard deviation.

## 3. Results

### 3.1. Screening of the Best PRL Concentration

PRL plays a key role in the process of lactation. Therefore, to select the best concentration, different concentrations of PRL were tested in the mammary epithelial cells of cows in this study. Mammary epithelial cells were treated with PRL at concentrations of 0 μg/mL, 0.5 μg/mL, 1 μg/mL, 2 μg/mL and 4 μg/mL. The levels of TAG concentration are shown in Figure 1A. PRL at concentrations of 0.5 μg/mL (1.20 ± 0.050, *p* = 0.011), 1 μg/mL (1.42 ± 0.070, *p* = 0.001), 2 μg/mL (1.51 ± 0.065, *p* < 0.001) and 4 μg/mL (1.49 ± 0.081, *p* = 0.001) increased TAG levels compared to NC. Casein beta and CSN2 are both the description for the same gene. Casein beta is a marker gene for milk fat metabolism in CMECs. *CSN2* (*β-casein*) and protein levels (Figure 1C) were determined. The results showed that the optimum concentration of PRL was 2 μg/mL. 

### 3.2. Molecular Mechanism of PRL Regulating miR-183 Expression in Cow Mammary Epithelial Cells 

The related indicators of mammary epithelial cells treated with PRL at different concentrations and times. The results showed that the expression of miR-183 was dependent on PRL at a certain concentration and time (Figure 2A,B). Meanwhile, there was a CpG island in the promoter region of the miR-183 5′-flanking region, which inhibited the expression of miR-183 after methylation (Figure 2C). 

To confirm the regulation of PRL to miR-183 related to this methylation site, mammary epithelial cells were treated with methylation inhibitors (5-aza-2′-deoxycitidine) compared with the control group (Untreated cells). PRL (0.55 ± 0.043) and 5-aza-dc + PRL (1.18 ± 0.044) were subsequently selected for the experiment. The results showed that 5-aza-dc (1.28 ± 0.047) could partially inhibit the PRL-mediated downregulation of miR-183 compared with PRL with 5-aza-dc + PRL (*p* < 0.001) (Figure 2D). Moreover, methylation levels of the NC, PRL and PRL + 5-aza-dc groups were detected by the hydrogen sulphite DNA sequencing method, and 5-aza-dc treatment was found to inhibit PRL-induced DNA methylation (Figure 2E). 

It was verified how PRL regulates the expression of miR-183 due to DNA methylation, and then how PRL modifies miR-183 should be clarified. It was hypothesized that induction of DNMT1 (DNA (cytosine-5-)-methyltransferase 1) activation by CMECs under PRL stimulation resulted in DNA methylation of the miR-183 5′-flanking region in the CpG island region, resulting in decreased expression of miR-183 at the transcriptional level. To test this hypothesis, two studies were conducted: (1) Detection of expression of DNMT1 at mRNA and protein levels after PRL stimulation. Studies have shown a significant increase in DNMT1 expression (1.48 ± 0.049, *p* = 0.001) under stimulation of PRL (Figure 3A,B). (2) Endogenous DNMT1 was interfered with siRNA, and expression of miR-183 was detected under PRL treatment. PRL (1.11 ± 0.083) could partially inhibit the downregulation (*p* < 0.001) of si-DNMT1 (1.67 ± 0.070) on miR-183 (Figure 3C). The above results showed that PRL down-regulates miR-183 by methylation in mammary epithelial cells (*p* < 0.05).

### 3.3. Transfection Efficiency of miRNA and siRNA

Figure 4A shows that the expression level of the miR-183 mimic in mammary epithelial cells was 60-fold higher than in the control group (NC-mimic, negative control mimic). Compared with the control group (NC-inhibitor, negative control-inhibitor), the expression level of the miR-183 inhibitor group was decreased by 80% after processing. The transfection efficiency of sirna-irs1 and sirna-dnmnt1 was determined by qRT-PCR for their respective miRNA levels. As shown in Figure 4B, in comparison with the control group, the expression level in the mammary epithelial cells treated with sirna-irs1 was downregulated by 74%, and the expression level of sirna-dnmnt1 was decreased by 68%. Furthermore, the protein expression level obtained from Western blots in mammary epithelial cells treated with sirna-irs1 was significantly downregulated compared with the control group (Figure 4C). The expression level of sirna-dnmnt1 was significantly downregulated compared to the control group (Figure 4D). The results indicated that the transfection efficiency of siRNA was good and could be used in the experiment.

### 3.4. miR-183 Specifically Targets IRS1 in CMECs

The current results showed that IRS1 mRNA expression was upregulated by miR-183 inhibition, but IRS1 gene expression was downregulated (*p* = 0.001) by miR-183 overexpression (0.70 ± 0.001) (Figure 5A). Furthermore, as shown in Figure 5C, the 3′-UTR of IRS1 binds to the miR-183 site. The results from luciferase reporter assays showed that overexpression of miR-183 downregulated (*p* = 0.003) the activity of the wild-type IRS1 gene 3′-UTR (0.41 ± 0.003) (Figure 5B). Moreover, the protein expression level of IRS1 was consistent with the trend of mRNA expression under the overexpression of miR-183 (Figure 5D).

The mRNA expression levels of miR-183 and IRS1 at different lactation stages and tissues of cows were determined in the experiment (Figure 6). The results showed that IRS1 was not only highly expressed during the peak period of lactation but was also highly expressed in mammary tissue. The expression level of miR-183 was opposite to that of IRS1. miR-183 likely binds to the 3′-UTR site of the IRS1 gene and negatively regulates the expression of mRNA and protein levels of the IRS1 gene.

### 3.5. Functions of miR-183 and IRS1 on CMECs

In this study, the lipid droplets and TAG in mammary epithelial cells under the overexpression and inhibition of miR-183 were assessed. Compared with the control group, the content of TAG and cholesterol level was decreased to 70% and 31% after treatment with miR-183, respectively. In addition, under the inhibition of miR-183, the content of TAG and cholesterol was significantly upregulated by 1.45 times and 1.58 times, respectively (Figure 7A,B). Additionally, the expression of CSN2 was reduced under miR-183 treatment (Figure 7C,D).

The results showed that miR-183 inhibited milk fat metabolism in mammary epithelial cells. miR-183 expression levels were detected in different tissues (Figure 6A) and lactation periods of cows (Figure 6B). The results showed that the expression level of miR-183 in mammary tissue was high, and it was lower during early lactation.

In ruminants, several genes may coordinate the regulation of milk fat and milk protein metabolism [21,22,23]. To understand the function of miR-183 in lipid metabolism, the mRNA levels of multiple genes related to fat metabolism were determined. The results showed that overexpression of miR-183 downregulated the mRNA expression levels of *SCD* (0.51 ± 0.103, *p* = 0.019), *ABCG1* (0.39 ± 0.002, *p* < 0.001), *FASN* (0.70 ± 0.070, *p* = 0.001) and *ABCA1* (0.60 ± 0.002, *p* = 0.001) (Figure 8A–D). In contrast, mRNA expression levels of *HSL* (1.41 ± 0.004) and *CPT1* (1.71 ± 0.063) were significantly upregulated (*p* < 0.001) when miR-183 was overexpressed (Figure 8E,F).

### 3.6. IRS1 Stimulated the Level of TAG and Increased the Accumulation of Lipid Droplets in CMECs

As shown in Figure 9A, the content of TAG was downregulated to 51% with IRS1 siRNA treatment compared to the control group in mammary epithelial cells. After IRS1 siRNA treatment, the cholesterol level was downregulated to 53% (Figure 9B). In addition, the mRNA and protein levels of CSN2 and the marker genes of milk fat metabolism were reduced after IRS1 siRNA treatment (Figure 9C,D).

### 3.7. siRNA-IRS1 Partially Remedy the Effect of miR-183 on TAG

Previous results found that miR-183 inhibits the synthesis of TAG. In contrast, the target gene of miR-183 (IRS1) promotes TAG synthesis. A “rescue” trial was designed to demonstrate that miR-183 performs its function through IRS1. The results showed that siRNA-IRS1 (1.10 ± 0.049) reduced (*p* < 0.001) the content of TAG in mammary epithelial cells with the treatment of inhibitor-miR-183 (1.52 ± 0.063). Therefore, the partial reduction in TAG content is likely due to sirna-irs1 (Figure 9E).

## 4. Discussion

### 4.1. Functions of miR-183 on CMECs

microRNAs (miRNA) are a class of endogenous non-coding RNA with a length of about 22 nucleotides (nt) and a high interspecies conservation [24]. Previous studies have reported that the human genome can transcribe thousands of miRNAs, and more than 60% of protein-coding genes are regulated by miRNA. Furthermore, miRNAs are regulated by epigenetic mechanisms. The study of the biological functions of miRNA is of specific significance for disease control and understanding biological processes [25,26]. Previous studies have shown that miR-183 plays an important role in the regulation of fetal development, glucose uptake, blood pressure regulation and cancer development [27]. Chen at al. 2012 [28] found that miR-183 was differentially expressed in the back fat of Large White pigs and Chinese Meishan pigs. Further studies showed that miR-183 expression was upregulated during the differentiation of 3T3-L1 cells and could directly bind to the 3′-UTR region of LRP6 to inhibit the activation of the classical Wnt signaling pathway, thereby promoting the differentiation of mouse preadipocytes [29]. However, evidence of the effects of miR-183 in cow mammary epithelial cells is still limited. This study showed the effect of miR-183 on the function of dairy cow mammary epithelial cells, which was explored through overexpression and inhibition. The current also demonstrated the important role of miR-183 in TAG synthesis in CMECs.

### 4.2. miR-183 Targets the IRS1 3′-UTR Directly

IRS1 is one of the target genes of miR-183, and IRS1 plays an important role in the regulation of milk fat metabolism [30,31]. In addition, insulin activation, carbohydrate and lipid metabolism are mediated and regulated by the IRS1 gene [32,33]. Therefore, IRS1 was selected for functional verification. Some nuclear receptor cofactors have an effect on the regulation of gene expression [34]. Insulin mediates a variety of metabolic responses, including the upregulation of glucose, fatty acids and gene-regulated responses. Phosphorylation of the insulin receptor causes activation of the insulin receptor, which transmits either negative or positive signals within the cell [35]. Previs et al. 2000 [36] reported that the IRS plays an important role in insulin-mediated carbohydrate and lipid metabolism. The binding sites of miR-183 and IRS1 3′-UTR were mapped. To further confirm that IRS1 is the target gene of miR-183, the luciferase double reporter system method was used for validation. Reporter gene analysis system is a simple method to study gene regulation, which can shift spontaneously under host cell gene expression regulation and further visually “reporting” the signal cascade reaction related to gene expression regulation in cells [37]. Segments of IRS1 3′-UTR for miR-183 were also cloned and examined for whether miR-183 potentially effects the activity of IRS1. The results showed that miR-183 significantly inhibited the activity of wild-type IRS1 3′-UTR. Results of a subsequent “rescue” test showed that sirna-irs1 could partially inhibit the effect of inhibitor-miR-183 on the regulation of TAG levels. From the functional point of view, IRS1 is the target gene of miR-183. 

### 4.3. PRL Regulates miR-183 Expression by Methylation Regulation

PRL is an important hormone in initiating and maintaining the physiological process of lactation [38,39]. It acts as regulatory role in DNA methylation. Gangisetty et al. (2015) [40] found that PRL could enhance methylation modification on the D2R gene promoter in human fetal breast and prostate tissues, affecting the expression level of D2R and changing the physiological function of the body]. Milk traits may be regulated by using DNA methylation and histone modification to interfere with the expression of galactolipid-related genes [41]. Chen et al. 2018 [20] reported that PRL could regulate the expression of DNMT1 and affected the body function through the apparent modification, which is consistent with our results. PRL has a certain regulatory effect on genomic DNA methylation, but information on the regulation effect of PRL on breast fatty acids is still limited. In the current study, the results suggest that there is a CpG island in the 5′-flanking promoter area of miR-183, which may inhibit the expression of miR-183 after methylation. Studies have shown that epigenetics regulates miRNA expression mainly through DNA methylation and histone modification [42,43]. microRNAs regulate gene expression by regulating DNA methylation transferase, maintaining DNA methylation level in cells or altering histone modification. Thus, epigenetics can cause significant changes in miRNA expression profiles. Preliminary studies showed that PRL stimulation could promote the expression of DNMT1 in dairy cow mammary epithelial cells and inhibited the expression of miR-183. 

## 5. Conclusions

Based on the above results, the promoted expression of DNMT1 in mammary epithelial cells led to DNA methylation modification in the 5′-lateral CpG island of miR-183 and inhibited the transcriptional expression of miR-183 under the stimulation of PRL. The downregulation of miR-183 leads to the upregulation of the expression of target gene IRS1, which in turn leads to changes in fatty acid metabolism as illustrated (Figure 10).

## Figures and Tables

**Figure 1 genes-11-00196-f001:**
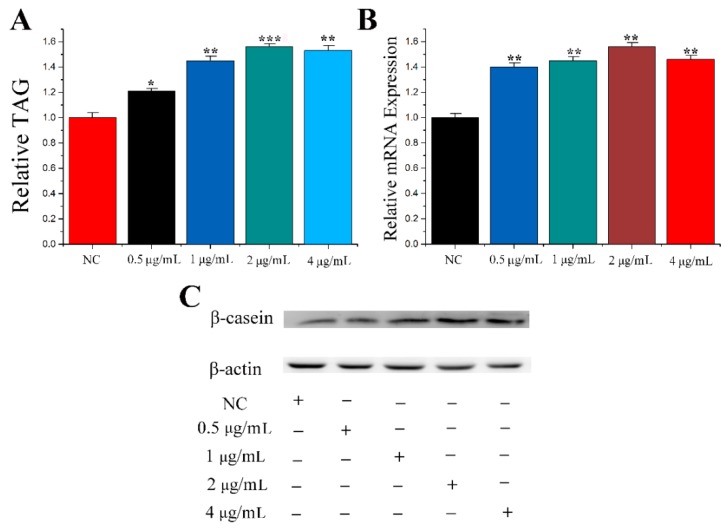
Screening for optimal concentration of PRL. (**A**) TAG levels in cells treated by PRL with different concentration. (**B**) The mRNA expression level of *β-casein*. (**C**) Western blot analyses of the expression of *β-casein* in the PRL treatment experiments. The effect of PRL on *β-casein* protein expression was evaluated by Western blot in CMECs. Values are presented as means ± standard deviation, * *p* < 0.05; ** *p* < 0.01; *** *p* < 0.001.

**Figure 2 genes-11-00196-f002:**
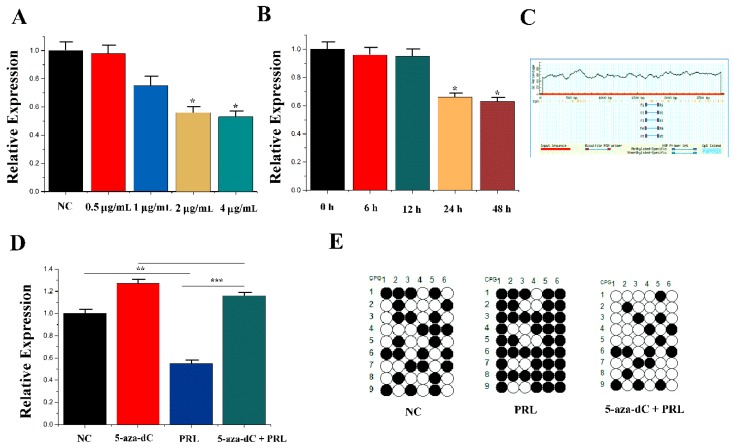
*PRL* affecting miR-183 expression. (**A**) The expression of miR-183 in CMEC treated with PRL for 48 h. (**B**) The expression of miR-183 after incubation with 2 μg/mL PRL. (**C**) CpG islands are present in the miR-183 promoter region. (**D**) The miR-183 expression levels in CMECs transfected with NC, 5-aza-Dc, PRL, and PRL +5-aza-dC. (**E**) Bisulfite DNA sequencing of CMECs transfected with NC, PRL, and PRL + 5-aza-dC. Values are presented as means ± standard deviation, * *p* < 0.05; ** *p* < 0.01; *** *p* < 0.001.

**Figure 3 genes-11-00196-f003:**
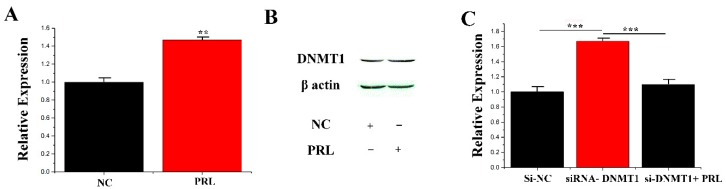
*DNMT1* affecting miR-183 expression. (**A**) The mRNA expression of DNMT1 was quantified. (**B**) Western blot of DNMT1 expression in the NC or PRL experiments. (**C**) The miR-183 expression level in CMECs. Values are presented as means ± standard deviation, ** *p* < 0.01; *** *p* < 0.001.

**Figure 4 genes-11-00196-f004:**
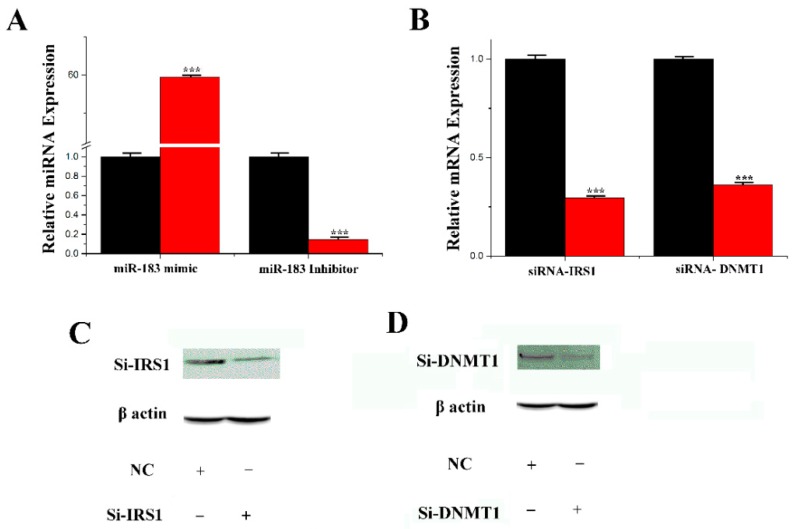
The effect of miR-183 and siRNA. (**A**) miR-183 expression levels in cells transfect with miR-183 mimic or inhibitor; miR-183 expression levels in transfected cells were compared with that of miR-183 controls (*n* = 6). Red bars represent the negative control; black bars represent miR-183 mimic or inhibitor; (**B**) The expression of IRS1 or DNMT1 was quantified by RT-qPCR (*n* = 6); (**C**,**D**): Si-IRS1 or DNMT1 protein expression was evaluated by Western blot in CMECs. Values are presented as means ± standard deviation, *** *p* < 0.001.

**Figure 5 genes-11-00196-f005:**
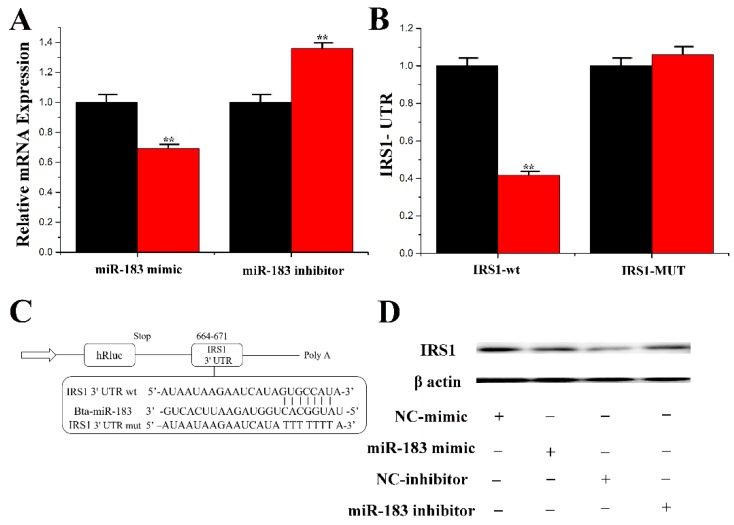
miR-183 specifically targets *IRS1.* (**A**) RT-qPCR quantification of the IRS1 expression (n = 6). Red bars represent the negative control; black bars represent miR-183 mimic or inhibitor; (**B**,**C**) Target site of miR-183 in IRS1 3′-UTR and the construction of the luciferase (Luc) expression vector fused with the IRS1 3′-UTR. WT represents the Luc reporter vector with the WT IRS1 3′-UTR (664-671); MU represents the Luc reporter vector with the mutation at the miR-183 site in IRS1 3′-UTR; (**D**) The effect of miR-183 mimicking and inhibiting IRS1 protein expression was evaluated by Western blot analysis in CMECs. All experiments were duplicated and repeated three times. Values are presented as means ± standard deviation, ** *p* < 0.01.

**Figure 6 genes-11-00196-f006:**
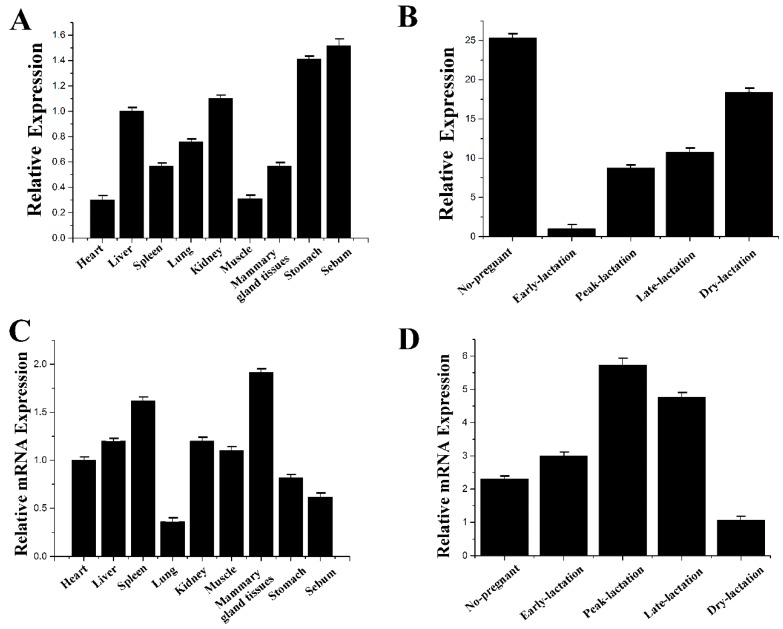
The expression level of miR-183 and *IRS1.* (**A**) miR-183 expression in various tissues of milk cows. (**B**) miR-183 expression in various periods of milk cows. (**C**) IRS1 expression in various tissues of milk cows. (**D**) IRS1 expression in various periods of milk cows. All experiments were duplicated and repeated three times. Values are presented as means ± standard deviation.

**Figure 7 genes-11-00196-f007:**
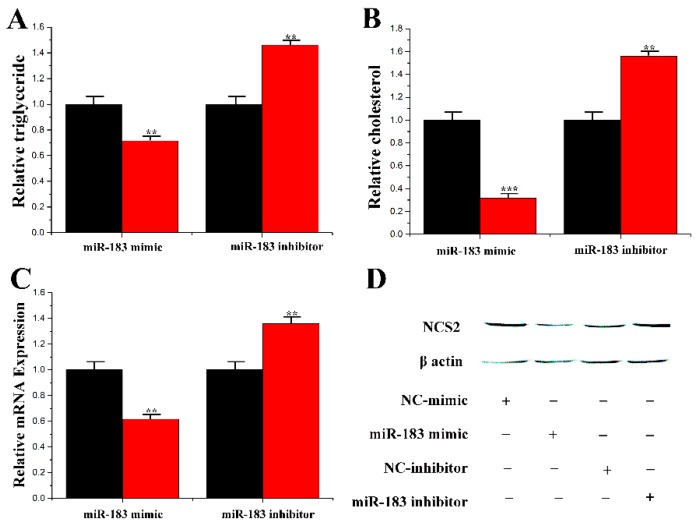
Functional evaluation of miR-183. (**A**) TAG levels in cells transfect with miR-183 mimic or inhibitor; TAG levels in transfected cells were compared with that of miR-183 controls (*n* = 6). Red bars represent the negative control; black bars represent miR-183 mimic or inhibitor; (**B**) Cholesterol levels in transfected cells were compared with that of control (*n* = 6). Red bars represent the negative control; black bars represent miR-183 mimic or inhibitor; (**C**) The expression of *β-casein* was quantified by RT-qPCR (*n* = 6); (**D**) The effect of miR-183 mimic or inhibitor on *β-casein* protein expression was evaluated by Western blot in CMECs. Total protein was harvested 48h post-treatment, respectively. All experiments were duplicated and repeated three times. Values are presented as means ± standard deviation, ** *p* < 0.01; *** *p* < 0.001.

**Figure 8 genes-11-00196-f008:**
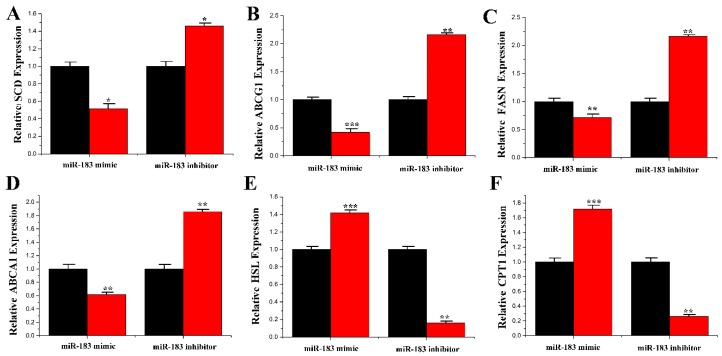
Expression of fatty acid metabolism-related genes. (**A**) The mRNA expression of *SCD*; (**B**) The mRNA expression of *ABCG1*; (**C**) The mRNA expression of *FASN*; (**D**) The mRNA expression of *ABCA1*; (**E**) The mRNA expression of *HSL*; (**F**) The mRNA expression of *CPT1*. The mRNA expression of *SCD*, *ABCG1*, *FASN*, *ABCA1*, *HSL* and *CPT1* were quantified by RT-qPCR (*n* = 6). Red bars represent the miR-183 mimic; black bars represent the miR-183 inhibitor; All experiments were duplicated and repeated three times. Values are presented as means ± standard deviation, * *p* < 0.05; ** *p* < 0.01; *** *p* < 0.001.

**Figure 9 genes-11-00196-f009:**
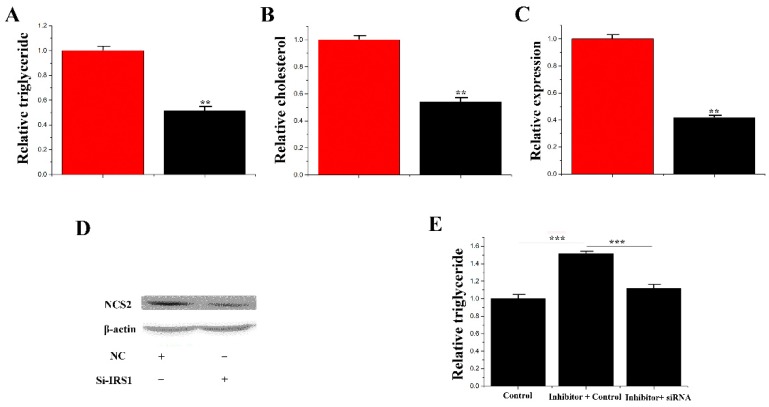
Functional evaluation of *IRS1*. (**A**) TAG levels in cells transfected with Si-NC or SiRNA- IRS1; TAG levels in transfected cells were compared with Si-NC (*n* = 6). Red bars represent the negative control; black bars represent SiRNA- IRS1; (**B**) Cholesterol levels in transfected cells were compared with controls (*n* = 6). Red bars represent the negative control; black bars represent the siRNA- IRS1; (**C**) The expression of *β-casein* was quantified by RT-qPCR (*n* = 6); (**D**) The effect of si-NC (60 nM) or siRNA- IRS1 (60 nM) on *β-casein* protein expression was evaluated by Western blot in CMECs; (**E**) TAG levels in cells transfected with control inhibitor (50 nM) + control siRNA (50 nM), inhibitor 183 (50 nM) + control siRNA (50 nM) and inhibitor 183 (50 nM) + siRNA- IRS1 (50 nM); TAG levels were compared with that of controls (*n* = 6). Values are presented as means ± deviation, ** *p* < 0.01; *** *p* < 0.001.

**Figure 10 genes-11-00196-f010:**
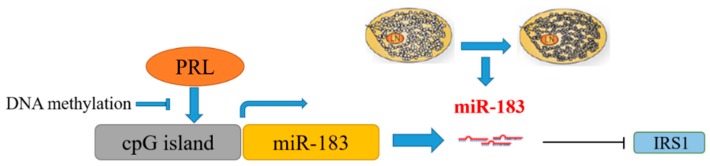
Mechanism of the PRL/miR-183/IRS1 pathway regulating milk fat metabolism in CMECs.

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
