# Peer review of "PRL/microRNA-183/IRS1 Pathway Regulates Milk Fat Metabolism in Cow Mammary Epithelial Cells"

_genes, 2020, doi:10.3390/genes11020196_

Round 1
Reviewer 1 Report
I think that the study can be of interest for the readers and can really contribute to a best knowledge of the mechanisms regulating the mammary synthesis of milk components.
The paper is complex, but well written; there are only few points to modify. Then I suggest publication after a minor revision of the following points:
- Be more precise on the editorial rules of the journal “Genes”: in particular, before citations in the text in square brackets you must put a space. Like this: “…protein metabolism [12].” and not like this: “…protein metabolism[12].”. When you cite a paper in this way “Avril et al. sequenced…. etc etc [16]”, it is better to put the number just after the citation, in this way “Avril et al. [16] sequenced…. etc etc” (see lines 50, 304, 316, 329, 333). Moreover, supplementary tables must be reported also in a paragraph at the end of the paper, just before the Author Contributions, as in the example that I report below:
Supplementary Materials: The following are available online at www.mdpi.com/xxx/s1. Table S1: The sequences used for primer design. Table S2: The PCR products sequencing data of nine reference genes. Table S3: Pairwise variation (Vn/n+1) analysis of nine candidate reference genes calculated by geNorm. Figure S2: Melting curves of the nine candidate reference genes.
- In a paper with such a complexity is very important to clearly define the meaning of the acronyms and the abbreviations used in the text. Also if the paper is directed to researchers expert in the field, sometimes the single elements are very specific of a particular pathway, so it is better that they are clearly defined. I mean, for example, in page 1 line 37, you introduce for the first time the miR-183, that is the protagonist of your paper, without explanation. You define it only in page 2 lines 44-49. This is only an example, but be careful to clearly define every name when you introduce it for the first time (LRP, GMEC, DNMT1, CMEC… and so on).
- Page 2 line 51: “Results showed some miRNA expression levels…” it is better to change to “Results showed that some miRNA expression levels…”.
- Page 2 line 67: it is better to write “5-aza” instead of “5aza”, as in lines 157 and 159.
- Material and Methods: this is the most important comment to make to the paper. In all material and Methods you always cite as reference the instructions of the kits, but never an international paper to support the analytical methods that you have used for your study. Please, if it’s possible, add some reference for the methods.
- Page 3 lines 109-112. “We constructed the luciferase 109 reporter vector with the IRS1 gene 3' -UTR wild type and its mutant vector (Table S4) as construction of target gene 3' -UTR luciferase reporter vector requires a series of experiments to verify the interference.” This sentence is not syntactically correct; please, rephrase it.
- Page 3 line 114: “Seventy-five microlitres of stop reagent was added to plates” must be changed to “Seventy-five microlitres of stop reagent were added to plates”.
- Page 3 line 120: “Four hundred and fifty ng genomic DNA was purified and recovered…” must be changed to “Four hundred and fifty ng genomic DNA were purified and recovered…”
- Page 4 line 135: For the readers that are not specific of this pathway, it should better to explain why do you involve the comparison with beta-casein, particularly, and not other caseins.
- Figure 2A legend: The expression of miR-183 in CMEC treated with different prolactin concentrations. At which time are measured? You should add this information in the legend.
- Figure 2B legend: The expression of 164 miR-183 after incubation with prolactin for different times. At which prolactin concentration are measured? You should add this information in the legend.
- Figure 6: The writings under the figures are too small and cannot be read. I would suggest dividing into two figures, with largest writings, one relating to the different tissues (A and C together) and one relating to the different periods (B and D together). Consequently they would become figures 6 and 7, and, obviously, the numbers of the subsequent figures would also have to be scaled.
- In the legends of the figures, sometimes the letters are written like this "A: etc. etc." and sometimes like this "(A) etc. etc. " (Figure 7 and Figure 9, for example). Look carefully at the editorial rules of the journal and standardize this writing in the legends.
- Page 10 line 308: The sentence “However, evident on effects of miR-183 in primary mammary epithelial cells of dairy cows is still limited.” must be changed to “However, evidence on effects of miR-183 in primary mammary epithelial cells of dairy cows is still limited.”
- Page 11 line 311: The sentence “The current also demonstrated an important role of miR-183 in TAG synthesis in CMECs.” must be changed to “The current study also demonstrated an important role of miR-183 in TAG synthesis in CMECs.”
- Figure 10 legend: “PRL/miR.183/ IRS1”. It is better to write it without the space before IRS1, like this: “PRL/miR.183/IRS1”; the same also in the title of the paper.
Author Response
Reviewer #1
General Comments:
I think that the study can be of interest for the readers and can really contribute to a best knowledge of the mechanisms regulating the mammary synthesis of milk components.
The paper is complex, but well written; there are only few points to modify. Then I suggest publication after a minor revision of the following points:
Response: Thank you very much for your valuable comments. Revision was made according to your suggestions.
Specific Comments:
Be more precise on the editorial rules of the journal “Genes”: in particular, before citations in the text in square brackets you must put a space. Like this: “…protein metabolism [12].” and not like this: “…protein metabolism[12].”. When you cite a paper in this way “Avril et al. sequenced…. etc etc [16]”, it is better to put the number just after the citation, in this way “Avril et al. [16] sequenced…. etc etc” (see lines 50, 304, 316, 329, 333).
Response: Revised.
- Moreover, supplementary tables must be reported also in a paragraph at the end of the paper, just before the Author Contributions, as in the example that I report below: Supplementary Materials: The following are available online at www.mdpi.com/xxx/s1. Table S1: The sequences used for primer design. Table S2: The PCR products sequencing data of nine reference genes. Table S3: Pairwise variation (Vn/n+1) analysis of nine candidate reference genes calculated by geNorm. Figure S2: Melting curves of the nine candidate reference genes.
Response: Thanks for the question raised, we have carried out modifications in the manuscript. We have modified as follows:
Supplementary Materials: The following are available online at www.mdpi.com/xxx/s1. Table S1: The sequences used for miR-183. Table S2: The PCR products sequencing data of nine reference genes. Table S3: The sequences used for IRS1-siRNA. Table S4: The sequences used for construction target gene 3' -UTR luciferase reporter vector.
- In a paper with such a complexity is very important to clearly define the meaning of the acronyms and the abbreviations used in the text. Also if the paper is directed to researchers expert in the field, sometimes the single elements are very specific of a particular pathway, so it is better that they are clearly defined. I mean, for example, in page 1 line 37, you introduce for the first time the miR-183, that is the protagonist of your paper, without explanation. You define it only in page 2 lines 44-49. This is only an example, but be careful to clearly define every name when you introduce it for the first time (LRP, GMEC, DNMT1, CMEC… and so on).
Response: All the abbreviations were defined in the manuscript.
- Page 2 line 51: “Results showed some miRNA expression levels…” it is better to change to “Results showed that some miRNA expression levels…”.
Response: Changed.
- Page 2 line 67: it is better to write “5-aza” instead of “5aza”, as in lines 157 and 159.
Response: Revised.
- Material and Methods: this is the most important comment to make to the paper. In all material and Methods you always cite as reference the instructions of the kits, but never an international paper to support the analytical methods that you have used for your study. Please, if it’s possible, add some reference for the methods.
Response: Some references were added according to reviewer’s suggestion.
- Page 3 lines 109-112. “We constructed the luciferase reporter vector with the IRS1 gene 3' -UTR wild type and its mutant vector (Table S4) as construction of target gene 3' -UTR luciferase reporter vector requires a series of experiments to verify the interference.” This sentence is not syntactically correct; please, rephrase it.
Response: The sentence was changed to “We separately constructed IRS1 gene 3' -UTR wild type and mutant vectors (Table S4), and then vectors were transferred into luciferase report vector to test activity.”
- Page 3 line 114: “Seventy-five microlitres of stop reagent was added to plates” must be changed to “Seventy-five microlitres of stop reagent were added to plates”.
Response: Changed.
- Page 3 line 120: “Four hundred and fifty ng genomic DNA was purified and recovered…” must be changed to “Four hundred and fifty ng genomic DNA were purified and recovered…”
Response: Revised.
- Page 4 line 135: For the readers that are not specific of this pathway, it should better to explain why do you involve the comparison with beta-casein, particularly, and not other caseins.
Response: Thanks for the suggestion. Casein beta and CSN2 are both the description for the same gene. Casein beta is a marker gene for milk fat metabolism in CMECs. Related research has been done in other articles.
As follows:
Chen, Z., Chu, S., Wang, X., Sun, Y., Xu, T., Mao, Y., Loor, J.J., and Yang, Z. (2019). MiR-16a Regulates Milk Fat Metabolism by Targeting Large Tumor Suppressor Kinase 1 (LATS1) in Bovine Mammary Epithelial Cells. J Agric Food Chem 67, 11167-11178.
Chen, Z., Qiu, H., Ma, L., Luo, J., Sun, S., Kang, K., Gou, D., and Loor, J.J. (2016). miR-30e-5p and miR-15a Synergistically Regulate Fatty Acid Metabolism in Goat Mammary Epithelial Cells via LRP6 and YAP1. Int J Mol Sci 17.
We also have specified the method to measure β-casein.
Protein abundance of β-casein in 293A cells (As a control group) or cow mammary epithelial cells (CMEC).
- Figure 2A legend: The expression of miR-183 in CMEC treated with different prolactin concentrations. At which time are measured? You should add this information in the legend.
Response: The information was added in the legend.
- Figure 2B legend: The expression of miR-183 after incubation with prolactin for different times. At which prolactin concentration are measured? You should add this information in the legend.
Response: We have carried out modifications in the article. We chose 2μg/mL of PRL-treated cells.
- Figure 6: The writings under the figures are too small and cannot be read. I would suggest dividing into two figures, with largest writings, one relating to the different tissues (A and C together) and one relating to the different periods (B and D together). Consequently they would become figures 6 and 7, and, obviously, the numbers of the subsequent figures would also have to be scaled.
Response: The Fig-6 was changed and it should be clear enough for the readers now.
- In the legends of the figures, sometimes the letters are written like this "A: etc. etc." and sometimes like this "(A) etc. etc. " (Figure 7 and Figure 9, for example). Look carefully at the editorial rules of the journal and standardize this writing in the legends.
Response: Thank you for the details. We have carried out modifications in the article.
- Page 10 line 308: The sentence “However, evident on effects of miR-183 in primary mammary epithelial cells of dairy cows is still limited.” must be changed to “However, evidence on effects of miR-183 in primary mammary epithelial cells of dairy cows is still limited.”
Response: Revised.
- Page 11 line 311: The sentence “The current also demonstrated an important role of miR-183 in TAG synthesis in CMECs.” must be changed to “The current study also demonstrated an important role of miR-183 in TAG synthesis in CMECs.”
Response: Revised.
- Figure 10 legend: “PRL/miR.183/ IRS1”. It is better to write it without the space before IRS1, like this: “PRL/miR.183/IRS1”; the same also in the title of the paper.
Response: Revised.

Reviewer 2 Report
The work presented in this manuscript seems novel and may be of interest, especially if it may help to improve the efficiency of milk production. However, in my opinion, it is not an easy paper to understand for the reader in the way it is now presented and a review of the writeing would be recommended.
Below I present my general comments, which are also detailed in the file I attach:
Material and Methods:
In my opinion, Statistical analysis should be much more detailed.Results
Significant P-values should be included in the text. When different samples are analyzed, and results are significant, please provide the exact value+SD in the text. I suggest a table with all the values analyzed.Main concern: There is a mix of information between different sections, above all Material/Methods and results (Detailed in the attached file). Please rewrite and set the information in the section that belongs.

Author Response
Reviewer #2:
General Comments:
The work presented in this manuscript seems novel and may be of interest, especially if it may help to improve the efficiency of milk production. However, in my opinion, it is not an easy paper to understand for the reader in the way it is now presented and a review of the writeing would be recommended.
Response: Thank you for your valuable advice. Revision was made according to your suggestions.
Below I present my general comments, which are also detailed in the file I attach:
Material and Methods:
In my opinion, Statistical analysis should be much more detailed.
Results
Significant P-values should be included in the text. When different samples are analyzed, and results are significant, please provide the exact value+SD in the text. I suggest a table with all the values analyzed.
Response: Thanks for the suggestions. The p-value was added in the significant results in the manuscript. All the figures in the manuscript provide sufficient information for readers to understand the trial. In addition, considering the length of the manuscript and the format of published papers in Genes, we don’t think it is a good idea to add exact value+SD and extra tables which provide the same information to the figures. (Please check results expression in recent publications of Genes: 1. Phylogenetic Analysis and Substitution Rate Estimation of Colonial Volvocine Algae Based on Mitochondrial Genomes doi:10.3390/genes11010115; 2. SNP Diversity in CD14 Gene Promoter Suggests Adaptation Footprints in Trypanosome Tolerant N’Dama (Bos taurus) but not in Susceptible White Fulani (Bos indicus) Cattle. doi:10.3390/genes11010112 )
Main concern: There is a mix of information between different sections, above all Material/Methods and results (Detailed in the attached file). Please rewrite and set the information in the section that belongs.
Specific Comments:
Line 30: “milk yield and quality” should be OK.
Line 33: Gland was added.
Line 38: Gland was added.
Line 39: “microRNA” was changed to “MicroRNA”.
Line 50: The year was added.
Line 57: There is no problem with the sentence of “important candidate miRNAs”. Thus, we didn’t change candidate to candidates.
Line 57-60: The sentence was changed to “Therefore, the main purpose of this study was to select differentially expressed miR-183 under PRL treatment and explore the function of miR-183 in isolated mammary epithelial cells.”
Line 63: Holstein cows was added.
Line 133: The different concentrations of prolactin were introduced in line 134 and 135.
Line 139-145: Revised.
Line 149: The sentence was changed to “The related indicators of mammary epithelial cells treated with PRL at different concentrations and times.”
Line 155: Revised.
Line 165: Revised.
Line 170: Revised.
Line 171: The “?” was removed.
Line 175: p-value was added.
Line 179: p-value was added.
Line 181-186: Revised.
Line 188: Revised.
Line 190: p-value was added.
Line 198: p-value was added.
Line 200: Transformation efficiency of siRNA was compared between the treatment group and NC group. The efficiency in the article is good. So we write "high.". We have replaced "high" with "good".
Line 206-208: Revised.
Line 210-212: The sentence was moved to M&Ms.
Line 212-215: The sentence was moved to discussion.
Line 218-220: The sentence was moved to M&Ms.
Line 225-226: It showed the relative expression levels of various tissues and organs. No specific tissue was used as the control group. It would be significantly different if the lowest expression level used as control.
Line 238-247: Revised.
Line 299: Revised.
Line 304: The year was added.
Line 307: Wnt signaling pathway is an important signal transduction system regulated by Wnt gene.
Line 321: Ref was added.
Line 328: Revised.
Line 330: Specie? It is human fetal breast
Line 335-336: The sentence was changed to “PRL has a certain regulatory effect on genomic DNA methylation, but the information on the regulation effect of PRL on breast fatty acids is still limited.”
Line 349: According to the published articles, it’s not a problem to put “(fig-10)” in conclusion.
Line 351: All the abbreviations were added in the manuscript.

Round 2
Reviewer 2 Report
In the light of the authors' review of the changes in the manuscript entitled "PRL / microRNA-183 / IRS1 pathway regulates milk fat metabolism in mammary epithelial cells of dairy cow," I think it is becoming more understandable today.
However, I still miss the following points:
I still think that the statistics section of the M & Ms could be explained more widely. Currently, it has been expanded with the explanation of how the results are provided (line 136-137) ... in my opinion, at present this is not entirely "true", since you do not provide the exact data. The results are reported only through the figures, but not in exact values. Si us plau, considerin el següent punt.
Include the numerical values +/- SD in the results section and the specific P values of the results (not the P of significance). I suggest you include them in the text. Thus, future studies will be able to compare the results much more precisely.
Author Response
Response letter
Journal: Genes
Manuscript ID: genes-706748
Title: PRL/microRNA-183/ IRS1 pathway regulates milk fat metabolism in mammary epithelial cells of dairy cow
Article type: Article
Reviewer 2
I still think that the statistics section of the M & Ms could be explained more widely. Currently, it has been expanded with the explanation of how the results are provided (line 136-137) ... in my opinion, at present this is not entirely "true", since you do not provide the exact data. The results are reported only through the figures, but not in exact values. Si us plau, considerin el següent punt.
Response: Thanks for the valuable suggestions. Some detail information was added in the statistics section of M & Ms.
In the manuscript, the measured data are relative values, because it needs to be unified. The details are as follows:
In order to visually exhibit the stability of each candidate gene were used to process the raw Ct values obtained by qRT-PCR. Data for the geNorm and NormFinder algorithms had to be processed by the formula: 2-ΔCt (ΔCt = each corresponding Ct value -the minimum Ct value). Then, by importing 2-ΔCt values into the programs, the stability parameters of each gene could be obtained. On the other hand, the best keeper is used for unification. BestKeeper (UXT) uses a different mechanism to choose the most stable genes. It is intuitive to select the internal reference genes more accurately under different external environments. The results also represent relative quantities. The relative concentrations of triglycerides and cholesterol were corrected by protein concentration (BCA protein detection) after detecting their absolute values. All the data shown in figures are relative values in this manuscript. In this way, the relative trend can be judged more accurately (suitable for comparison between two pairs). Please check our previous published papers (1. Pathological Features of Staphylococcus aureus Induced Mastitis in Dairy Cows and Isobaric-Tags-for- Relative-and-Absolute-Quantitation Proteomic Analyses. doi.org/10.1021/acs.jafc.7b05461; 2. miR-148a and miR-17-5p synergistically regulate milk TAG synthesis via PPARGC1A and PPARA in goat mammary epithelial cells. doi.org/10.1080/15476286.2016.1276149; 3. Mechanism of prolactin inhibition of miR‐135b via methylation in goat mammary epithelial cells. doi.org/10.1002/jcp.25925Include the numerical values +/- SD in the results section and the specific P values of the results (not the P of significance). I suggest you include them in the text. Thus, future studies will be able to compare the results much more precisely.
Response: Thanks for the suggestion. The numerical values +/- SD were added in the text.
